EMBO
Molecular Medicine

# The inwardly rectifying K⁺ channel KIR7.1 controls uterine excitability throughout pregnancy

Conor McCloskey[1], Cara Rada[2], Elizabeth Bailey[1], Samantha McCavera[1], Hugo A van den Berg[3], Jolene Atia[1], David A Rand[3], Anatoly Shmygol[1], Yi-Wah Chan[1], Siobhan Quenby[1], Jan J Brosens [1], Manu Vatish[1], Jie Zhang[1], Jerod S Denton[4], Michael J Taggart[5], Catherine Kettleborough[6], David Tickle[6], Jeff Jerman[6], Paul Wright[6], Timothy Dale[7], Srinivasan Kanumilli[7], Derek J Trezise[7], Steve Thornton[8], Pamela Brown[9], Roberto Catalano[9], Nan Lin[10], Sarah K England[2] & Andrew M Blanks[1,*]

## Abstract

Abnormal uterine activity in pregnancy causes a range of important clinical disorders, including preterm birth, dysfunctional labour and post-partum haemorrhage. Uterine contractile patterns are controlled by the generation of complex electrical signals at the myometrial smooth muscle plasma membrane. To identify novel targets to treat conditions associated with uterine dysfunction, we undertook a genome-wide screen of potassium channels that are enriched in myometrial smooth muscle. Computational modelling identified Kir7.1 as potentially important in regulating uterine excitability during pregnancy. We demonstrate Kir7.1 current hyper-polarizes uterine myocytes and promotes quiescence during gestation. Labour is associated with a decline, but not loss, of Kir7.1 expression. Knockdown of Kir7.1 by lentiviral expression of miRNA was sufficient to increase uterine contractile force and duration significantly. Conversely, overexpression of Kir7.1 inhibited uterine contractility. Finally, we demonstrate that the Kir7.1 inhibitor VU590 as well as novel derivative compounds induces profound, long-lasting contractions in mouse and human myometrium; the activity of these inhibitors exceeds that of other uterotonic drugs. We conclude Kir7.1 regulates the transition from quiescence to contractions in the pregnant uterus and may be a target for therapies to control uterine contractility.

**Keywords** pregnancy; parturition; potassium channels; uterus; myometrium
**Subject Categories** Pharmacology & Drug Discovery; Urogenital System

## Introduction

At the end of pregnancy, the quiescent uterus must become highly contractile to mediate parturition. While the mechanisms initiating parturition in mammals are diverse (Smith, 2007), a final common pathway of uterine transition to a contractile phenotype appears to converge on the expression of a group of proteins, including the oxytocin receptor, prostaglandin endoperoxidase synthase 2, ion channels and connexin 43, that alter the uterine myometrial smooth muscle (MSM) cell from a state of low intrinsic excitability and refractory to stimulation, to a state that has high intrinsic excitability and is susceptible to stimulation (Garfield *et al*, 1977; Fuchs *et al*, 1982; Slater *et al*, 1995; Garfield & Maner, 2007). Control of stimulation is itself subject to complex gene-environment interactions, which act either independently of, or complimentary to, the underlying physiological changes of the uterus (Cha *et al*, 2013).

Irrespective of the underlying stimulus, the regulation of the duration and frequency of the myometrial contraction critically depends on control of calcium entry through voltage-gated L- and T-type calcium channels (Amedee *et al*, 1987; Blanks *et al*, 2007). Calcium is not only an important second messenger in the generation of force via calcium-calmodulin-dependent myosin light chain kinase, but also depolarizes the plasma membrane, allowing for activation of other voltage-dependent ion channels (Word *et al*, 1994; Brainard *et al*, 2007). This voltage-mediated control of uterine excitability is modulated in a gestation-dependent manner in all mammalian species. In particular, mid-gestation is characterized by a hyperpolarized membrane potential close to the reversal potential for potassium $E_k$ (Casteels & Kuriyama, 1965). As pregnancy progresses towards term, the myometrium becomes increasingly

1    Division of Reproductive Health, Clinical Sciences Research Laboratories, Warwick Medical School, University of Warwick, Coventry, UK
2    Division of Basic Science Research, Department of Obstetrics and Gynecology, School of Medicine, Washington University in St. Louis, St. Louis, MO, USA
3    Warwick Systems Biology & Mathematics Institute, University of Warwick, Coventry, UK
4    Vanderbilt Institute of Chemical Biology, Vanderbilt Institute for Global Health, Vanderbilt University School of Medicine, Medical Center North, Nashville, TN, USA
5    Institute of Cellular Medicine, Newcastle University, Newcastle upon Tyne, UK
6    Centre for Therapeutics and Discovery,Medical Research Council Technologies, London, UK
7    BioPark, Essen BioScience Ltd, Welwyn Garden City, Hertfordshire, UK
8    Exeter Medical School, Exeter, UK
9    MRC Centre for Reproductive Health (CRH), Queen's Medical Research Institute, University of Edinburgh, Edinburgh, UK
10   Department of Mathematics, Washington University, St. Louis, MO, USA
    *Corresponding author. Tel: +44 2476968703; Fax: +44 2476968653; E-mail: andrew.blanks@warwick.ac.uk

depolarized, to approximately −45 mV at parturition (Casteels & Kuriyama, 1965; Parkington et al, 1999). The mechanism underpinning this crucial, evolutionarily conserved process remains unknown.

A number of potassium channels have been demonstrated to play a role in shaping the myometrial action potential and modulating myometrial contractility (Brainard et al, 2007). The putative roles for the different channels are diverse and depend on the physiological environment. These roles range from voltage-dependent modulation of the action potential waveform (Knock et al, 1999), to modulating responses to intracellular calcium release through $BK_{ca}$ and $SK_3$ (Khan et al, 1993; Pierce et al, 2008), intracellular ATP concentration through $K_{ATP}$ (Khan et al, 1998) and uterine stretch through tandem pore channels (Tichenor et al, 2005).

Within the potassium channel super-family, inwardly rectifying potassium channels represents good candidates for the regulation of resting membrane potential. Kir7.1 is a member of the inwardly rectifying potassium channel sub-family and is only 38% identical to its closest relative, Kir4.2 (Doring et al, 1998; Krapivinsky et al, 1998; Partiseti et al, 1998). Kir7.1 is expressed in visceral tissues and some neurones within the CNS (Krapivinsky et al, 1998; Nakamura et al, 1999; Ookata et al, 2000; Shimura et al, 2001), although little is known about its functional role. Studies in retinal pigment epithelial cells indicate that the channel may play an integral role in setting resting membrane potential and modulating $K^+$ recycling (Shimura et al, 2001). A rare genetic mutation in Kir7.1 causes autosomal-dominant snowflake vitreoretinal degeneration characterized by congenital degeneration of ocular tissues including the vitreous (Hejtmancik et al, 2008). Kir7.1 has also been linked to developmental pathways. For example, the jaguar/obelix mutation in zebra fish renders the Kir7.1 homologue non-functional. As a result, melanophores fail to respond appropriately to external cues, leading to melanosome aggregation and the phenotype of a broader striping pattern (Iwashita et al, 2006). Kir7.1 may also be involved in palate formation in mice. Kcnj13 was identified as one of 8 genes whose mis-expression correlates with formation of cleft palate in TGF3beta knockouts, though the precise role of the channel remains to be described (Ozturk et al, 2013).

In this study, we demonstrate the crucial importance of Kir7.1 in modulating uterine contractility in mice and humans. We show that the physiological function of Kir7.1 is to maintain a hyperpolarized membrane potential during uterine quiescence and that removal of this hyperpolarizing drive renders the uterus more excitable. Furthermore, we show that Kir7.1 also modulates the action potential waveform, modifying the excitation-contraction cycle by participating in key stages of repolarization. Pharmacological manipulation of this normal physiological process could be an alternative strategy to treat an atonic uterus and obstetric haemorrhage.

## Results

The aim of our study was to identify a potassium channel with the appropriate biophysical attributes to regulate the myometrial resting membrane potential during gestation. To identify candidates, we undertook a genome-wide qRT-PCR screen of all known $K^+$ channels and associated subunits in cDNA pools generated from laser-capture micro-dissected MSM and whole myometrial tissue. Of seven transcripts enriched in MSM (Supplementary Table S1), only KCNJ13 coded for a $K^+$ channel (Kir7.1) with appropriate biophysical attributes. Kcnj13 transcript levels increased markedly in the pregnant mouse uterus during mid-gestation, peaking on gestational day (GD) 15, and followed by a sharp decline towards term (C57BL/6J mice deliver in the morning of GD19; Fig 1A). KCNJ13 transcript levels (Fig 1B) were also significantly lower ($P < 0.05$) in samples taken from pregnant women in labour at term than in samples from women not in labour. Immunoblot of myometrial lysates from both labour and non-labour samples demonstrated a single immuno-reactive band at ~42 kDa. As positive and negative controls, we also tested human eye and adipose cell lysate, respectively (Fig 1C and Supplementary Fig S1). Furthermore, Kir7.1 immuno-reactivity was expressed in MSMs and was absent in the vasculature in both human and mouse myometrial samples (Fig 1D), supporting the specificity of the laser-capture screen.

To determine the functional significance of this $K^+$ channel in the uterus, we first investigated its electrophysiological properties in freshly dissociated mouse MSMs. Under voltage-clamp conditions, an inwardly rectifying potassium current (Fig 2A,B) was inhibited by VU590, a known Kir7.1 inhibitor (Lewis et al, 2009). Consistent with the finding that Kir7.1 expression was higher at GD15 than GD18, the VU590 sensitive current density at -150 mV was significantly greater ($P < 0.05$) on GD15 when compared to GD18 (Fig 2C).

To understand the function of Kir7.1 in the generation of the myometrial action potential, we modelled the potential impact of changes in Kir7.1 channel density on myometrial electrogenesis using a Hodgkin-Huxley type current summing model. In free-running simulations of membrane potential, increasing Kir7.1 channel density (within the range measured in our experimental data) hyperpolarized resting membrane potential and decreased calcium entry during the action potential (Fig 3). Furthermore, simulations predicted that overall membrane conductance during the excited phase of the action potential is so finely balanced that small changes in Kir7.1 current density exerted large effects on membrane potential. Given its expression profile and biophysical properties, we hypothesized that Kir7.1 is a key regulator of myometrial membrane potential during gestation.

To assess the role of Kir7.1 in regulating uterine contractility experimentally, we used lentiviral vectors expressing miRNA targeting Kcnj13 or the human Kir7.1 channel to inhibit and over-express the channel in murine MSM, both in vitro and in vivo. Knockdown of Kir7.1, in vitro, significantly increased the contractile activity integral (2.5-fold) (area under the time-force curve), duration (2.2-fold) and maximum force (1.3-fold) of phasic contractions when compared to scrambled miRNA control (Fig 4 and Supplementary Fig S2). Conversely, over-expression of Kir7.1 significantly decreased all three parameters. To determine whether these alterations in contractility were due to electrogenic effects, we used sharp microelectrodes to measure membrane potentials in the treated myometrial strips. Knockdown of Kir7.1 depolarized resting membrane potential with extended excited periods (Fig 5A,B), whereas over-expression of Kir7.1 hyperpolarized resting membrane potential and suppressed excitability (Fig 5C). Mean resting membrane potentials differed significantly between the two treatment groups (Fig 5D).

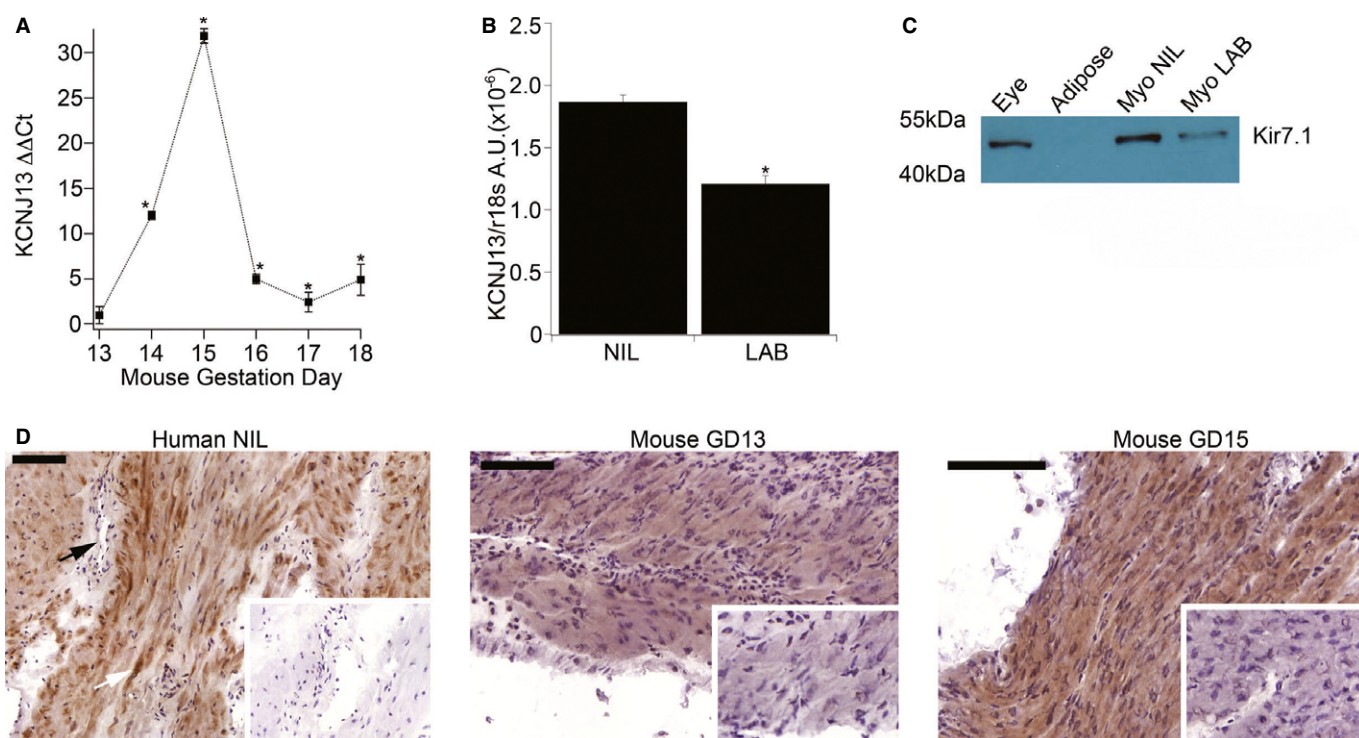

**Figure 1.  Kir7.1 is expressed in uterine myocytes and is regulated in pregnancy in mice and humans.**

A    mRNA expression of *Kcnj13* in mice (*n* = 5; mean ± SD, per GD) normalized to GD13. *$P < 0.05$, Student's *t*-test.
B    mRNA expression of *KCNJ13* (plotted as arbitrary units relative to 18s rRNA) in human myometrial samples from women at term not in labour (NIL) and at term in labour (LAB) (*n* = 8; mean ± SD, per group). *$P < 0.05$, Student's *t*-test.
C    Immunoblot of pooled lysates from eye, adipose tissue, NIL and LAB myometrium (*n* = 4) probed with antibody to human Kir7.1 (full blot available in Supplementary Fig S1).
D    Immunohistochemistry for Kir7.1 in human NIL myometrium (left panel), GD13 murine myometrium (centre panel) and GD15 murine myometrium (right panel). Arrow indicates absence of staining in blood vessel. Inset panels show tissue treated with pre-absorbed primary antibody control counterstained with haematoxylin. Scale bar = 100 μm.

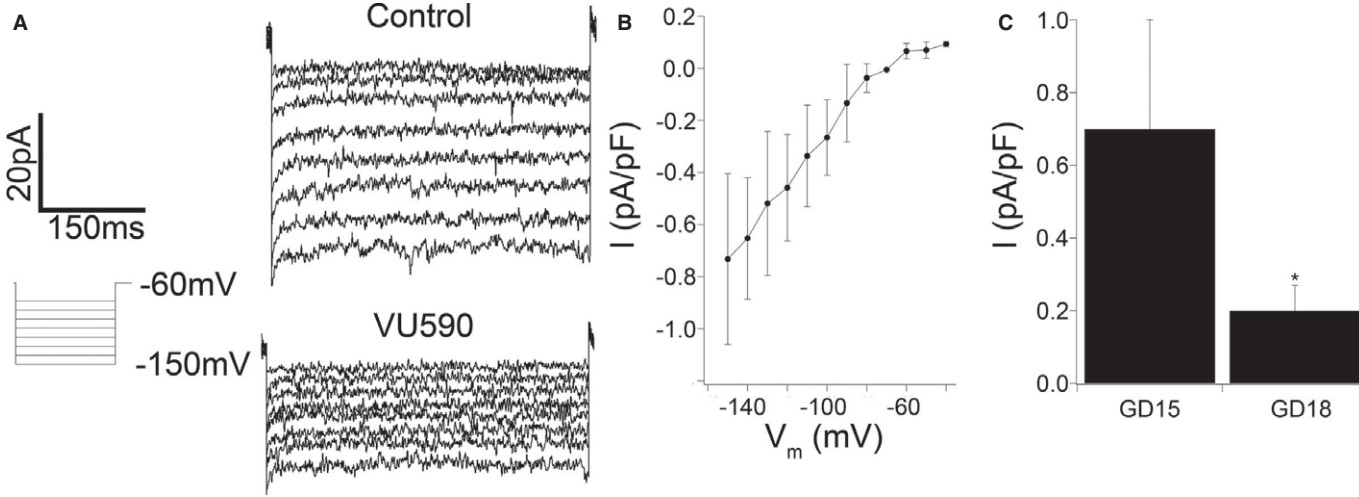

**Figure 2.  Kir7.1 current in uterine myocytes decreases from mid-pregnancy to term.**

A    Measurement of inwardly rectifying, VU590-sensitive current in freshly dissociated GD15 murine myometrial cells. Shown are voltage-clamp recordings in the presence and absence of 10 μM VU590.
B    Current-voltage relation (*n* = 5; mean ± SD, per data point) of current density [VU590 subtracted from control (vehicle alone)].
C    Current density (pA/pF) at −150 mV and 500 ms in freshly dissociated murine myometrial cells from GD15 and GD18 (*n* = 5 mean ± SD, per GD). *$P < 0.05$, Student's *t*-test.

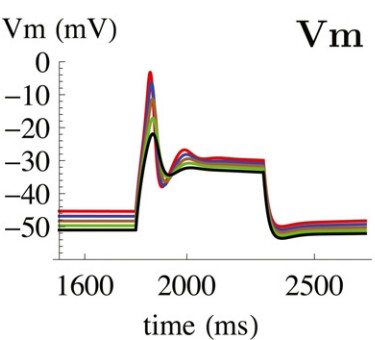

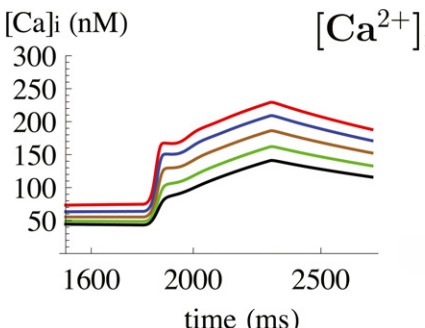

**Figure 3.  A free-running simulation of the effect on the myometrial action potential waveform of increasing densities of Kir7.1.**
Time-dependent effect of increasing Kir7.1 channel densities on $V_m$ (mV, left) and $[Ca]_i$ (nM, right). Increasing density of Kir7.1 within experimentally determined values hyperpolarizes resting membrane potential, whereas decreasing membrane excitability during depolarizing excursions in $V_m$ leading to decreased calcium entry.

Ion channel function *in vivo* may differ to that observed *in vitro,* because of either effects of the sample preparation, or influence of *in vivo* factors not captured *in vitro*. To account for these potential experimental confounders and to assess the phenotype of targeting Kir7.1 during pregnancy, we injected mice uteri on GD9 with anti-Kir7.1 miRNA or scrambled miRNA control. To record uterine activity, we surgically implanted a pressure catheter with remote telemeter into the injected horn of both knockdown and control mice. Intra-uterine pressure was recorded continuously, and the animals were monitored by video. Mice in which Kir7.1 was knocked down had significantly increased intra-uterine pressure when compared to control mice (Fig 6 and Supplementary Fig S3). Consistent with our findings that Kir7.1 expression and channel activity were higher at GD15 than GD18, the effect on intrauterine pressure was more pronounced in mid-gestation.

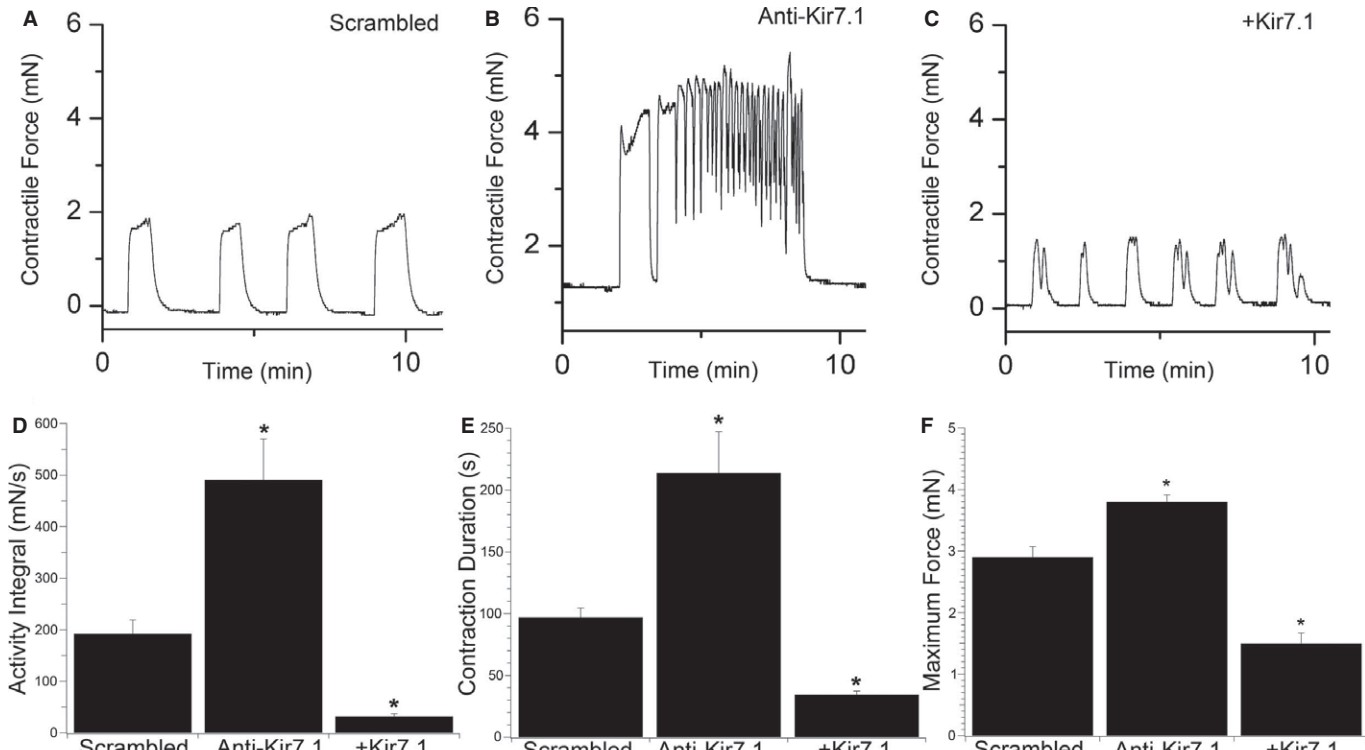

**Figure 4.  Knockdown of Kir7.1 *in vitro* increases myometrial activity and promotes tonic contractions.**
A–C   Representative time-force recordings of phasic contractions demonstrating (A) the effect of scrambled miRNA compared to (B) knockdown (Anti-Kir7.1) and (C) overexpression (+Kir7.1) of Kir7.1 on contractility in murine GD15 myometrial strips.
D–F   Mean data are summarized (*n* = 8; mean ± SD, per group of experiments) as activity integral (area under the time-force curve) (D), contraction duration (E) and maximum force (F). *$P < 0.05$, compared to scrambled control by ANOVA with Tukey's *post hoc* test.

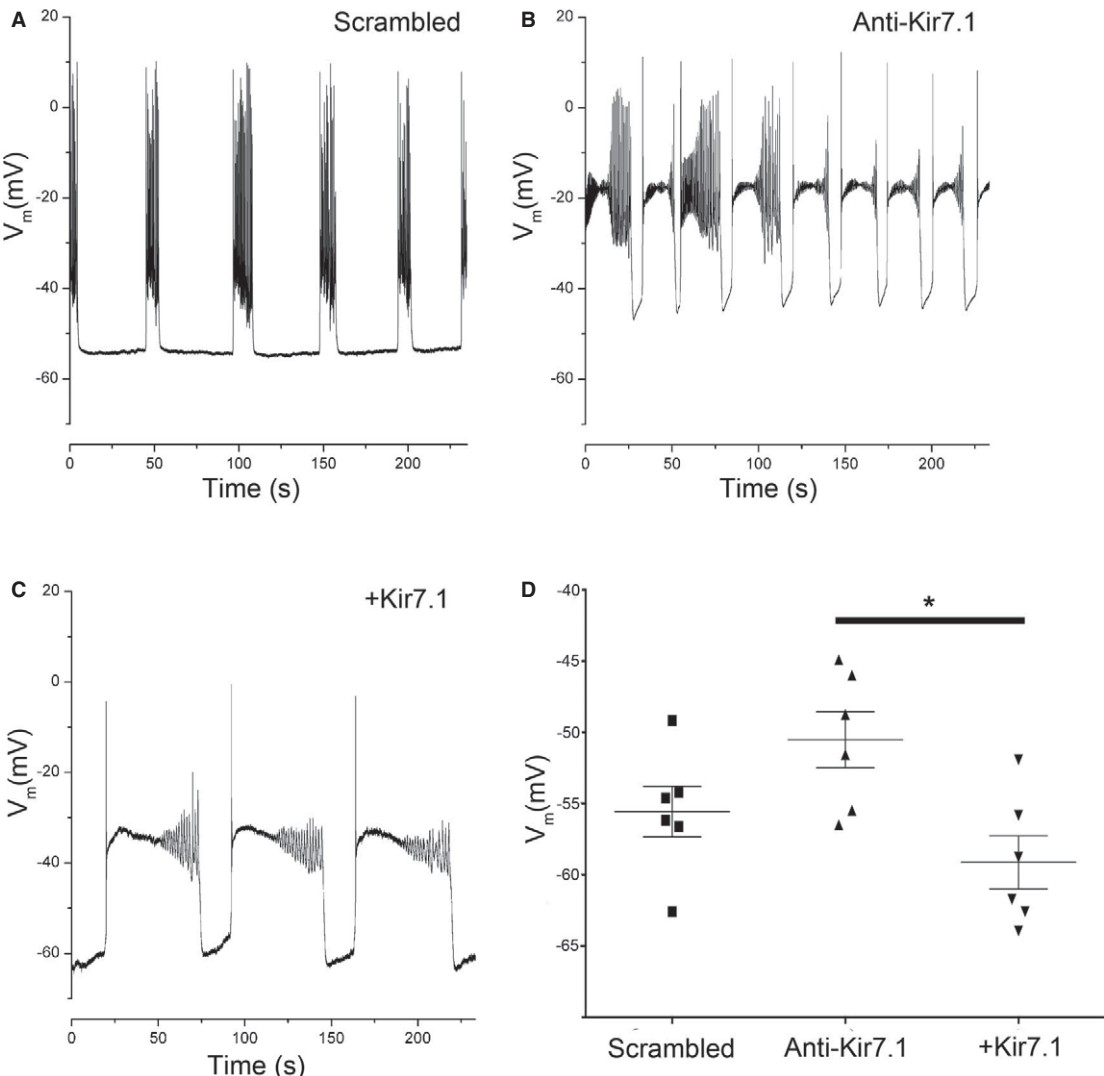

**Figure 5. Knockdown of Kir7.1 *in vitro* depolarizes resting membrane potential.**

A–C   Representative membrane potential recordings in current clamp configuration from murine myometrial strips (A) treated with scrambled control miRNA (B) treated with Kir7.1 knockdown (Anti-Kir7.1) and (C) overexpressing Kir7.1 (+Kir7.1).

D   Resting membrane potential in current clamp configuration from murine myometrial strips (*n* = 6; mean ± SD) from experiments depicted in (5A–C). *$P < 0.05$ by Student's *t*-test.

The increased contractility associated with Kir7.1 knockdown raised the possibility that pharmacological targeting of this ion channel could be of clinical value, for example in the management of severe post-partum haemorrhage. To assess this possibility, we first performed current clamp microelectrode recordings of spontaneous action potentials in isolated murine myometrial strips. We observed slow, inter-contraction, depolarization of resting membrane potential followed by transient complex action potentials (Fig 7). Administration of 10 μM VU590 rapidly depolarized resting membrane potential to threshold (Fig 7Ai), followed by a sustained plateau potential that was reversible on wash out (Fig 7Aii and Aiii). In myometrial strips from women, application of VU590 increased the activity integral, largely due to an increase in contraction duration (Fig 7B), with contractions lasting several hours observed.

To assess the potential therapeutic benefit of inhibiting Kir7.1, we compared the effect of VU590 with oxytocin, the established front-line treatment for post-partum haemorrhage. The effect of VU590 on contractile force in samples taken from GD15 and GD18 mice was dependent on dose and gestation (Fig 8A) and when used in combination with oxytocin, increased activity integral by 172 ± 14 fold and 90 ± 42 fold on GD15 and GD18, respectively, as compared to 4 ± 2 fold and 8 ± 3 fold for OXT treatment alone (Fig 8B). The effect of VU590 was similar in human term myometrium, which was also dose-dependent, with the observed increase in activity integral, largely due to an increase in contraction duration (Fig 8C,D). Importantly, sufficient channels remain at term to generate a significant phenotypic effect, underscoring the potential post-partum benefit of pharmacological intervention.

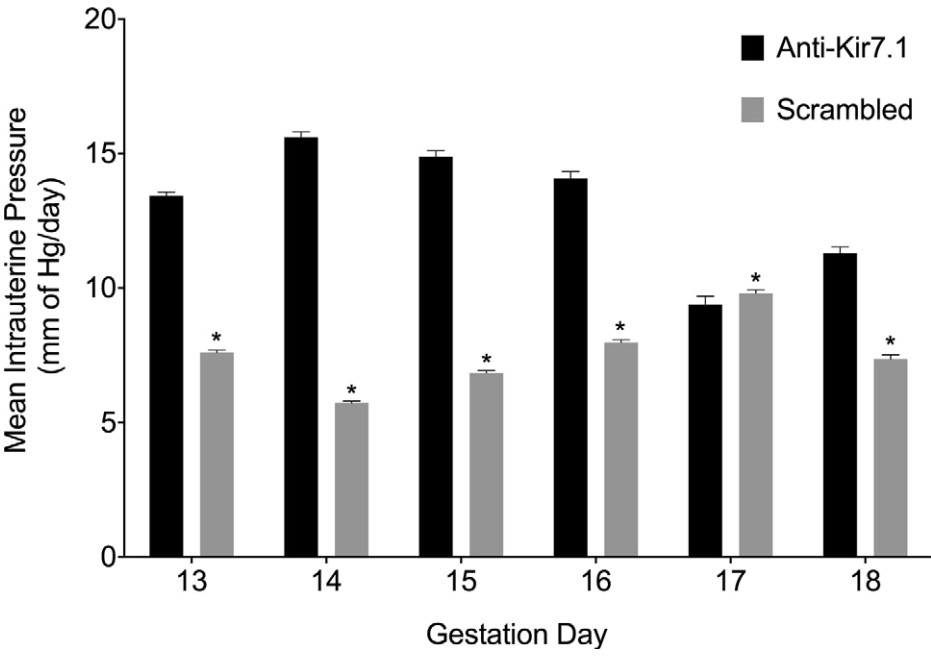

**Figure 6. Knockdown of Kir7.1 *in vivo* significantly increases intrauterine pressure.**
Mice in which Kir7.1 was knocked down (Anti-Kir7.1) had significantly increased intrauterine pressure when compared to mice injected with scrambled miRNA lentivirus (scrambled) from GD13 to GD18 (Fig 6 and Supplementary Fig S3). (*n* = 6 per time point). *$P < 0.05$; see methods for statistical model.

To expand the pharmacological tools for Kir7.1 inhibition, we used population patch-clamp technology to screen a library of known ion channel inhibitors in a Chinese Hamster Ovary cell line expressing Kir7.1 (Supplementary Fig S4 and Supplementary Methods). We identified compounds with different structures than VU590 to assess whether the phenotype of Kir7.1 inhibition was similar (Fig 9A). We also manufactured a control compound 1,3-Bis (5-nitro-1H-benzo[d]imidazol-2-yl)propane (BNBI), which is known to inhibit the structurally related Kir1.1 (Renal outer medulla potassium channel) but does not inhibit Kir7.1 (Bhave *et al*, 2011). Administration of 100 μM BNBI had no detectable effect on uterine contractility in the GD15 mouse (Fig 9B), suggesting that loss of efficacy for Kir7.1 inhibition is associated with loss of pro-contractile activity. Finally, administration of 10 μM MRT2000769, a potent Kir7.1 inhibitor that is structurally unrelated to VU590, induced long-lasting contractions similar to those observed with VU590 (Fig 9C). The effectiveness of MRT2000769 further supported the correlation between effective Kir7.1 inhibition and the phenotype of long-lasting contractions (Supplementary Fig S5).

## Discussion

Inwardly rectifying potassium channels are known to regulate diverse but important physiological processes such as insulin secretion in the pancreas, regulation of the cardiac action potential, parasympathetic stimulation and potassium reuptake in the kidney (Hibino *et al*, 2010). In most cases, the channels act to hyperpolarize resting membrane potential by remaining persistently open, allowing the efflux of potassium. In excitable tissues, this dampens electrical activity, while in epithelial cells, the potassium gradient

created in combination with energy-dependent ion pumps is used to transport other ions. In secretory cells, inhibition of the current causes depolarization and calcium entry leading to a secretion event (Liu *et al*, 2001; Ashcroft, 2005; Hebert *et al*, 2005). The important physiological roles of Kir channels are underscored by the many diseases that are associated with Kir channel malfunction, such as Bartter's syndrome, Anderson syndrome, short Q-T syndrome and neonatal diabetes (Derst *et al*, 1997; Andelfinger *et al*, 2002; Edghill *et al*, 2004; Priori *et al*, 2005; Ellard *et al*, 2007).

Here we present the novel finding that Kir7.1 is a crucial regulator of membrane potential in uterine myocytes during pregnancy in both mice and humans. In mid-gestation, high expression of Kir7.1 keeps the resting membrane potential close to the reversal potential for potassium, increasing the depolarizing drive required to initiate an action potential, calcium entry and subsequent contraction. At term, this damping of excitability is lost by reduction, but not complete loss, of Kir7.1. Our results also indicate that inhibition of Kir7.1 when combined with oxytocin administration is synergistic, a mechanism that may be related to the sensitivity of Kir7.1 to intracellular phosphatidylinositol 4,5-bisphosphate depletion (Pattnaik & Hughes, 2009). Such a mechanism, when acting in conjunction with a decreased gap junction density and decreased receptors to stimulatory ligands, could provide a robust means of maintaining uterine quiescence during gestation (Garfield *et al*, 1977; Fuchs *et al*, 1982; Smith, 2007).

Within the Kir superfamily, Kir7.1 displays several unique properties such as low sensitivity to $Ba^{2+}$ and $Cs^{+}$, low single channel conductance, no internal block by $Mg^{2+}$ ions, and a relative insensitivity to external $K^{+}$ concentration, tetraethylammonium ($IC_{50} > 10$ mM) or 4-aminopyridine ($IC_{50} \sim 10$ mM) (Krapivinsky *et al*, 1998). In *in vitro* expression systems, Kir7.1 exhibits rapid

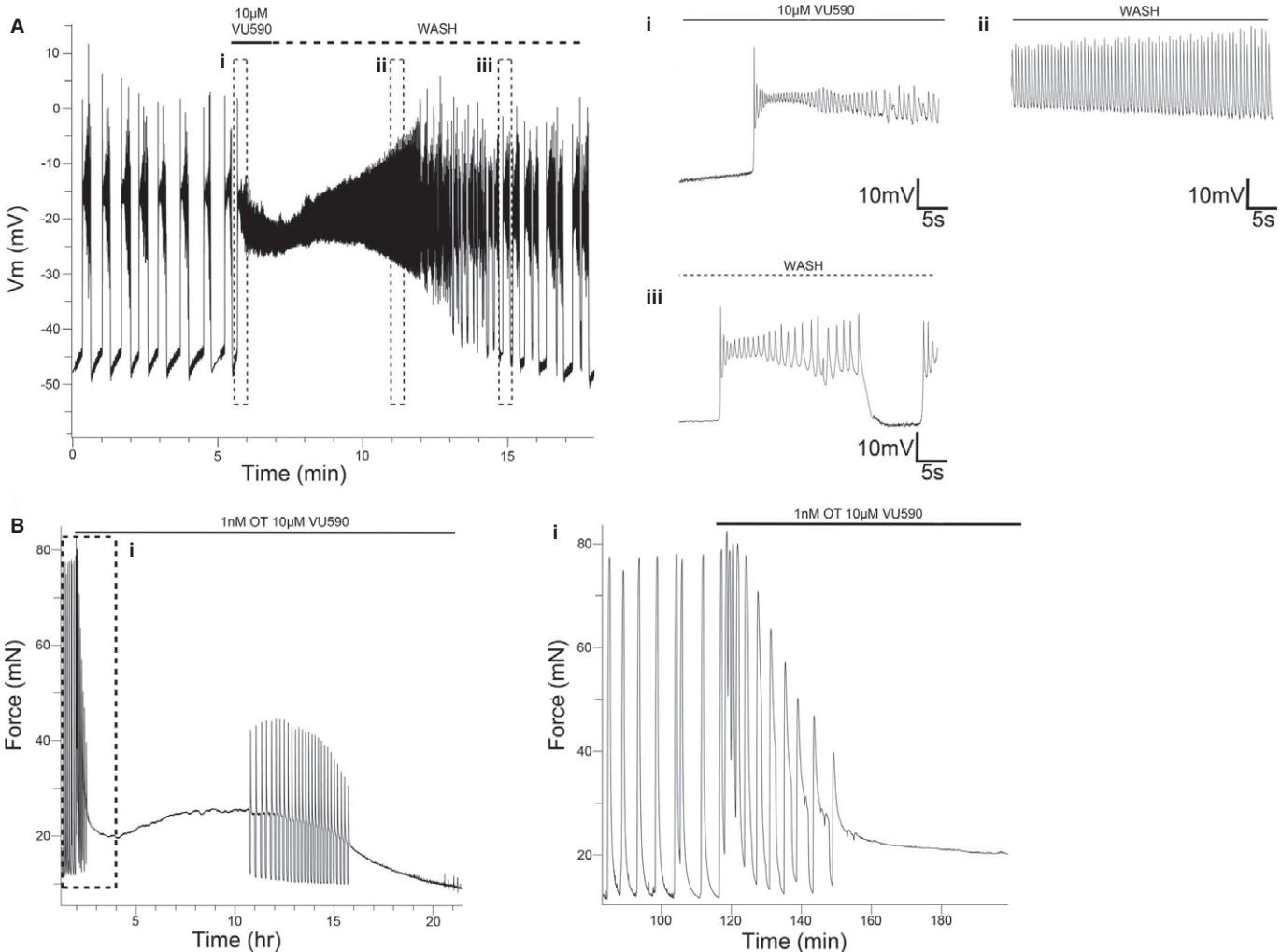

**Figure 7.   Pharmacological inhibition of Kir7.1 *in vitro* in human and murine myometrium induces membrane depolarization and long-lasting contractions.**

A    Representative membrane potential recording in current clamp configuration from a murine GD15 myometrial strip. (i) Addition of 10 μM VU590 depolarizes resting membrane potential (note slope change from resting potentials of preceding phasic bursts) and leads to a sustained plateau potential. (ii) Upon washout, spike potentials recover as plateau potential hyperpolarizes. (iii) On complete washout, phasic bursting resumes.

B    Addition of 1 nM oxytocin and 10 μM VU590 to human myometrial strips stimulates long-lasting contractions. (i) Initial component of the response is phasic, followed by establishment of a tonic contraction.

activation kinetics and is essentially non-inactivating at potentials negative to 40 mV with a small single channel conductance in both recombinant and native cells (Doring *et al*, 1998; Krapivinsky *et al*, 1998; Shimura *et al*, 2001). Since the channel has a low conductance and little voltage or time dependence in physiological ranges, high expression of this channel gives rise to a stable and hyperpolarized resting membrane potential. Thus, the biophysics of Kir7.1 is ideally suited to regulating myometrial smooth muscle cells during mid-gestation quiescence.

Transgenic mouse models have demonstrated the critical nature of control of myometrial membrane potential during parturition. For example, mice overexpressing the small conductance potassium channel SK3 suffer acute uterine dystocia, and both mother and pups die during delivery (Bond *et al*, 2000). Similarly, mice harbouring a smooth muscle specific deletion of the important uterine gap junction protein connexin43 demonstrate a significant delay

in delivery and increased mortality of pups (Doring *et al*, 2006). These data, in conjunction with our *in vivo* observations demonstrating a significant increase in intrauterine pressure in the absence of endocrine changes, underline the importance of the development of uterine excitability during gestation to the overall delivery process. It is notable that the decrease in Kir7.1 expression precedes progesterone withdrawal in the mouse suggesting that modulation of excitability is, at least in part, independent of progesterone. These biophysical factors act in addition to, and in concert with, other endocrine/paracrine changes that alter uterine stimulants (Cha *et al*, 2013) and lead to uterine disorders such as preterm labour or dysfunctional labour. The overall role of membrane potential in the control of parturition in the context of other controlling factors is summarized in Fig 10.

In addition to a clear effect on resting membrane potential, our computer modelling predicted that increasing Kir7.1 activity

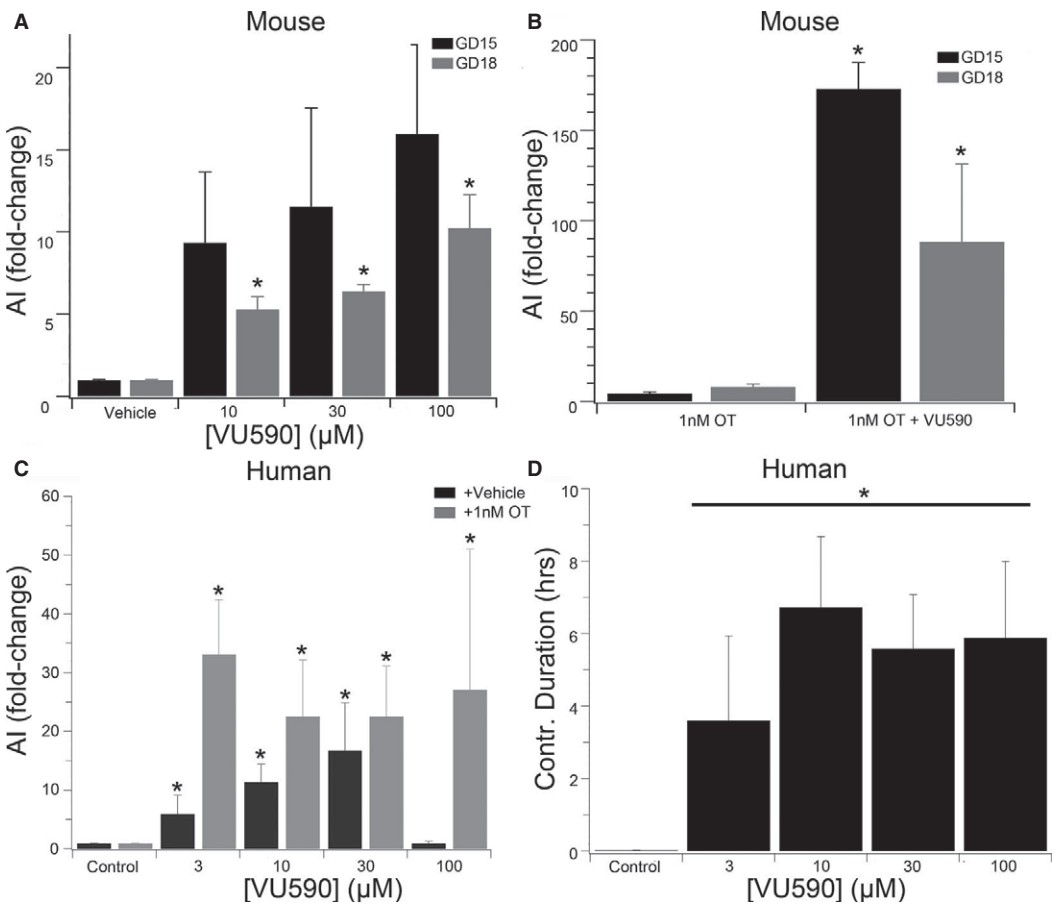

**Figure 8.  Pharmacological inhibition of Kir7.1 *in vitro* in human and murine myometrium stimulates longer-lasting contractions than oxytocin.**

A  Dose- and gestation-dependent effect of VU590 on murine myometrial contractility [$n = 5$, activity integral expressed as a fold-change of pre-treatment contractions (mean ± SD, per GD)]. *$P < 0.05$ GD18 vs GD15 per dose by ANOVA with Tukey's *post hoc* test.

B  As in (A) but comparing the effect of 1 nM oxytocin with 1 nM oxytocin + 10 μM VU590. *$P < 0.05$ OT vs OT = VU590 per GD by Student's *t*-test.

C  Dose-dependent effect of VU590 (Black Bars) and VU590 + 1 nM oxytocin (grey bars) on human myometrial contractility ($n = 8$; activity integral expressed as fold-change over pre-treatment contractions, mean ± SD). *$P < 0.05$ compared to control by ANOVA with Tukey's *post hoc* test.

D   Dose-dependent effect of VU590 on contraction duration in human myometrial strips taken at term ($n = 8$, mean ± SD). *$P < 0.05$ compared to control by ANOVA with Tukey's *post hoc* test.

modulates calcium entry during the plateau phase of the complex action potential, resulting in a decrease in contractile force and inefficient contractions. Decreasing Kir7.1 has the effect of stressing the repolarizing drive such that increased contributions from other $K^+$ channels are required to ensure action potential repolarization (Supplementary Fig S6) (Fink *et al*, 2006; Greenwood *et al*, 2009; McCallum *et al*, 2011). These effects explain the longer action potentials observed during pharmacological inhibition and experimentally induced reduction in the expression of Kir7.1 protein. When Kir7.1 current density is reduced to very low levels, the myometrial smooth muscle effectively loses phasic behaviour and manifests a new tonic-like tone. The greatly increased contraction length induced by Kir7.1 inhibition may be useful for reducing blood loss in an atonic uterus. As a treatment for post-partum haemorrhage, Kir7.1 block would have the advantage of being more potent than current first line treatments. The mechanism of action also circumvents the agonist pathways (Fig 10) targeted by current treatments that are prone to desensitization during failed labour inductions.

## Materials and Methods

### Ethical approval

All procedures involving women were conducted within the guidelines of The Declaration of Helsinki and were subject to local ethical approval (REC-05/Q2802/107). Prior to surgery, informed written consent for sample collection was obtained.

All animal procedures complied with the guidelines for the care and use of animals set forth by the National Institutes of Health. The Animal Studies Committee at Washington University in St. Louis approved all protocols (protocol number 20110138 to Sarah K. England). Adult C57BL/6J (Jackson Laboratory) female mice were mated at 8 weeks of age until 6 months of age. Mice were mated for 2-h time periods, and the presence of a copulatory plug was marked 0 days postcoitum (dpc). Animals were housed in Washington University School of Medicine vivarium in the BJCIH building, which is an AAALAC (Association for Assessment and Accreditation of Laboratory Animal Care International)

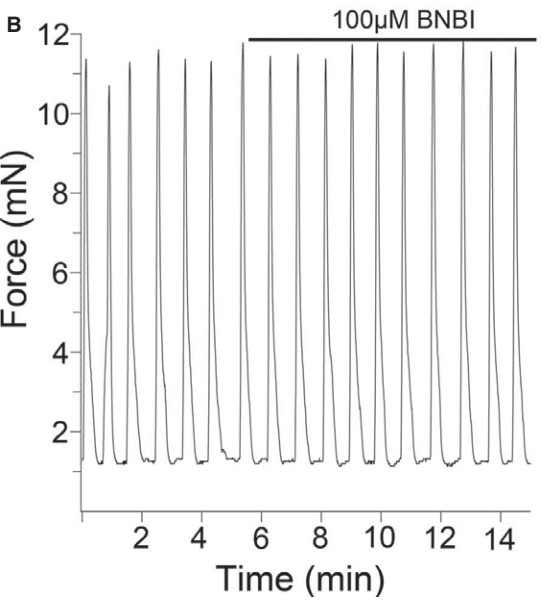

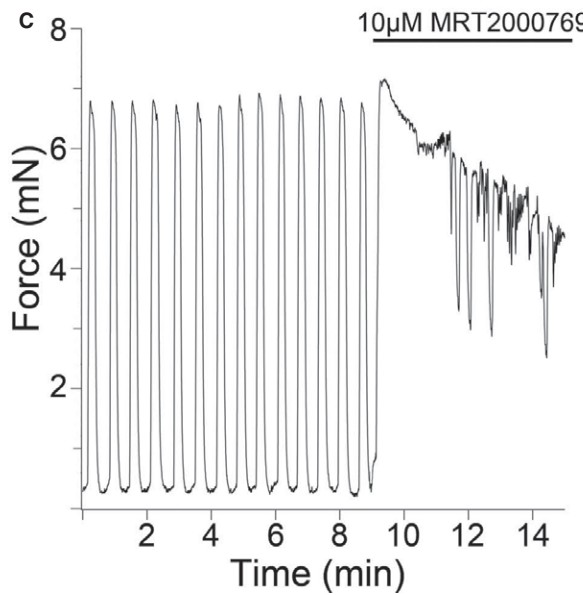

**Figure 9.  Stimulation of uterine contractions correlates with Kir7.1 inhibitory potency.**

A    The structures of the three compounds tested in this study. BNBI, a potent Kir1.1 inhibitor does not inhibit Kir7.1. MRT2000769 is structurally unrelated to VU590 and was identified as inhibiting Kir7.1 in a high throughput electrophysiology screen. VU590 is the first described inhibitor of Kir7.1.

B, C  Effects of BNBI (B) and MRT2000769 (C) on murine GD15 myometrial contractility.

approved animal care facility. For *in vivo* intrauterine pressure measurements, eight pregnant mice were injected with scrambled miRNA and eight pregnant mice were injected with Kir7.1 miRNA.

**Subject criteria and selection**

Subjects were recruited into two groups, spontaneous labour (LAB) and elective caesarean section not in labour (NIL) between 38 and 40 weeks gestation. The LAB group was undergoing caesarean section for reasons of undiagnosed breech. LAB was defined as regular contractions (< 3 min apart), membrane rupture and cervical dilatation (> 2 cm) with no augmentation.

**Sample collection**

At caesarean section, samples were collected before syntocin administration by knife biopsy from the lower uterine segment incision. Samples were washed briefly in saline and flash-frozen in liquid nitrogen for mRNA, immunoblot or immunohistochemistry

analyses. Samples for cell isolation or contractility experiments were placed in ice-cold modified Krebs–Henseleit (m-KHS) solution and used the same day.

*Laser capture screen*

mRNA was extracted from 100 mg of frozen human myometrium using Trizol reagent (Invitrogen), and further column purified by RNeasy kit (Qiagen) according to the manufacturer's instructions. Total RNA was quantified by spectrophotometer and further tested for quality and purity by bioanalyser (Agilent Technologies) according to the manufacturer's instructions. cDNA was generated from 100 ng of mRNA using Superscript III (Invitrogen) according to the manufacturer's instructions and stored at −80°C until qRT-PCR analysis.

Laser Capture Microdissection (LCM): All slides, LCM caps, Haematoxylin and Eosin (H&E) Staining Kit for LCM and solutions were obtained from Molecular Machines & Industries. Briefly, cryomold-mounted myometrial samples were transferred from −70°C and equilibrated in a pre-cooled cryostat (−30°C) for 10 min. Sections (8 µM) were cut and H&E stained according to

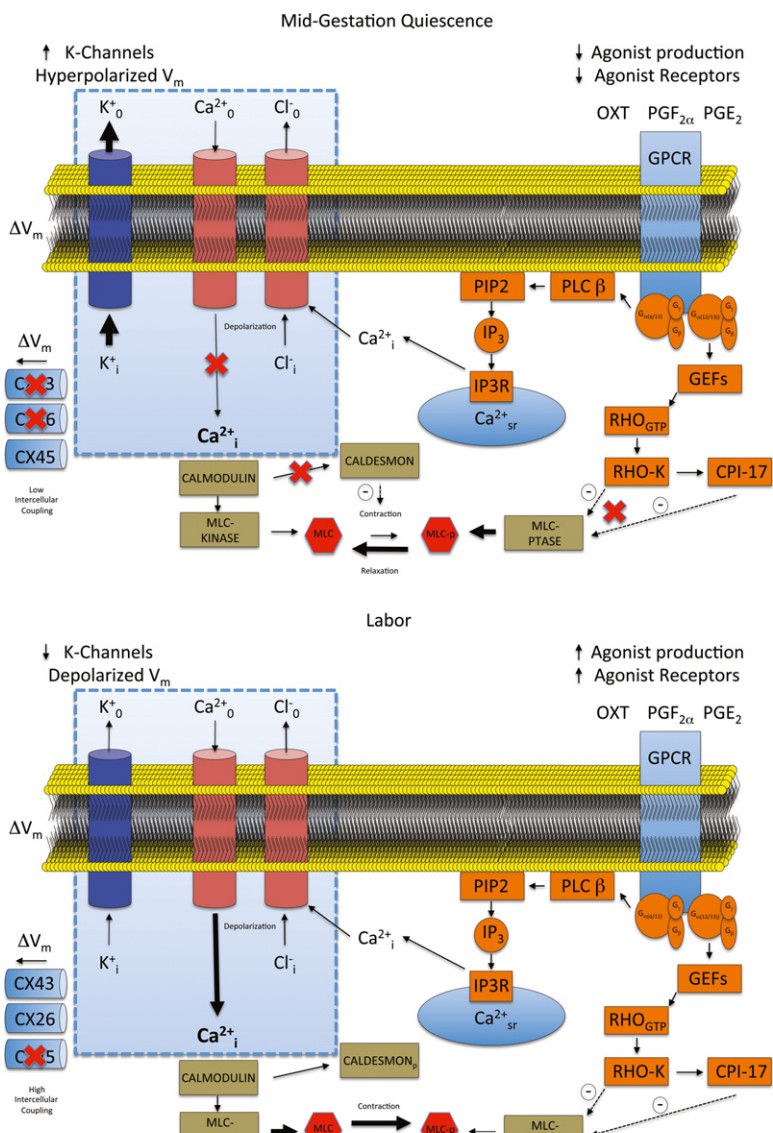

**Figure 10.   Model of the role of Kir7.1 in maintenance of uterine quiescence.**

In normal labour in the human myometrium, changes take place over a number of weeks that increase both the intrinsic electrical excitability of the cell (blue box) and, by altering cellular receptors and contractile machinery, the susceptibility to stimulation. Alterations in stimulation either by infection or gene-environment interaction can precipitate preterm labour (Cha *et al*, 2013); however, changes in control of the myometrial membrane potential or interference with functional gap junctions can affect labour even under normal endocrine conditions (Bond *et al*, 2000; Doring *et al*, 2006). In this study, we demonstrate that alteration of the function of a single potassium channel profoundly alters uterine contractility. Loss of Kir7.1 function depolarizes the plasma membrane and promotes voltage-gated calcium entry. In addition, the duration of the depolarization is extended, preventing the normal phasic contractions. In this way, the process of excitation-contraction coupling in uterine myocytes overrides pharmaco-contraction coupling. This effect could be used for therapeutic benefit. (Key: CPI-17 = C-kinase potentiated protein phosphatase-1 inhibitor. CX26 = Connexin 26. CX43 = Connexin 43. CX45 = Connexin 45. GEFs = Guanine nucleotide exchange factors. GPCR = G-protein coupled receptors. IP3 = inositol 1,4,5-trisphosphate. IP3R = inositol 1,4,5-trisphosphate receptor. MLC = Myosin light chain. MLCp = Phosphorylated myosin light chain. PIP2 = Phosphatidylinositol 4,5-bisphosphate. PLC β = Phospholipase C. RHOgtp = Ras homologue gene family, member A. RHO-K = Rho-associated, coiled-coil containing protein kinase 1.)

the manufacturer's instructions. Slides were dehydrated in graded alcohols and xylene and placed in a desiccator until laser capture. At least 50,000 $\mu m^2$ of human uterine myocytes was captured for each gene. mRNA was extracted from each isolation by using the Picopure RNA isolation kit (Arcturus) according to the manufacturer's instructions. Reverse transcription was as previously described. PCR primer design and qRT-PCR are described in the Supplementary Methods.

*Immunohistochemistry*

Frozen sections (8 $\mu M$) were fixed in ice-cold acetone for 5 min prior to incubation in phosphate-buffered saline (PBS). Endogenous peroxidase activity was blocked by immersing the slides for 30 min in freshly prepared 0.3% hydrogen peroxide in PBS. Slides were pre-incubated in 1.5% non-immune goat serum in PBS for 30 min at room temperature and then in primary anti-Kir7.1 antibody (1:100; Alomone) overnight at 4°C in a humidified chamber. Primary

antibody pre-absorbed for 24 h with recombinant antigen served as negative control. Staining was visualized using the Vectastain Elite ABC rabbit IgG kit (Vector Laboratories) according to the manufacturer's instructions.

### Immunoblot

After tissue was suspended in RIPA lysis buffer containing protease inhibitor cocktail tablets, mechanically homogenized and cleared by centrifugation, the supernatant was aliquoted and frozen. Whole tissue and protein lysates from adult human eye, adult mouse adipose and adult mouse brain were purchased from Novus Biologicals (Cambridge, UK) at a stock concentration of 5 mg/ml. All protein concentrations were confirmed with the BioRad assay (Bio-Rad laboratories, Hemel Hempstead, UK). A total of 40 μg of protein per sample (pooled from 4 biopsies each for NIL and LAB samples) was subjected to SDS-PAGE according to standard protocols. The membrane was blocked in 5% milk protein solution (Marvel, Lincs, UK) for 1 h at room temperature, incubated with primary rabbit polyclonal anti Kir7.1 antibody (1:200; Alomone Labs, Jerusalem) overnight at 4°C in blocking buffer and then incubated with polyclonal goat anti-rabbit HRP secondary antibody (1:100; Dako, Ely, UK). ECL Plus (GE Healthcare LTD, Amersham place, UK) was used to detect signal. To confirm equal loading, the blot was treated with Restore western blot stripping buffer (Thermo Fisher Scientific, Hemel Hempstead, UK) for 15 min at room temperature, washed, blocked and re-probed with an antibody to human β-actin.

### Electrophysiology

#### Cell isolation

Strips of myometrium from the longitudinal layer ($2 \times 2 \times 20$ mm) of time-mated C57BL/6J mice were isolated and washed in $Ca^{2+}$ and $Mg^{2+}$ free Hank's balanced salt solution (HBSS) at 37°C for 10, 20 and 30 min, respectively, followed by 45 min incubations in digestion solution (Roche Blendzyme 3) at 37°C according to the manufacturer's instructions. Digestion was terminated by several dilutions with fresh HBSS. Cells were dispersed by slow trituration through a wide-bore fire-polished glass pipette in HBSS. Single myometrial cells were filtered through a 200-μM gauze and stored in HBSS for use within six hours.

#### Voltage clamp

A drop of myometrial cell suspension was placed in a glass-bottomed Petri dish and mounted on the stage of an inverted microscope (IX51, Olympus). After settling (~ 10 min), cells were perfused with bath solution at a rate of 1–2 ml/min at 37°C. Patch pipettes were fabricated (Model P-87; Sutter Instruments, Novato, CA, USA) from 1.5-mm glass capillaries and had a resistance of 2.0–4.0 MΩ when filled with pipette solution (containing in mM: KCL 140; EGTA 1.1; $CaCl_2$ 0.06; Hepes 10; $MgCl_2$ 2; adjusted to pH 7.2 at 25°C with 5 M NaOH). Liquid junction potential was zeroed prior to seal formation. Transmembrane currents were recorded with an amplifier (Axopatch 700b; Axon Instruments) using the perforated patch configuration of the whole cell patch-clamp technique (Rae *et al*, 1991). The antibiotic amphotericin B (720 μg/ml) was used to perforate the cell membrane. Series resistance was compensated after membrane perforation. Currents were

elicited by stepping to a range of potentials between −150 mV and +80mV from a holding potential of −60 mV. To isolate currents that were sensitive to inhibition by drug application, difference currents were obtained by electronic subtraction of traces. Currents were filtered at 10 kHz and sampled at 5 kHz. Voltage protocols were delivered via a Digidata 1440a computer interface using pCLAMP 9.0 software (Molecular Devices, Sunnyvale, CA, USA).

#### Current clamp

Strips ($5 \times 10$ mm) of murine myometrium from the longitudinal layer were pinned out on a sylgard base and perfused with m-KHS containing (in mM: NaCl, 133; KCl, 4.7; Tes, 10; glucose, 11.1; $MgSO_4$, 1.2; $KH_2PO_4$, 1.2; $CaCl_2$, 2.5; adjusted to pH 7.4 at 25°C with NaOH) at 37°C on an upright microscope (MVX10, Olympus). Tissue was incubated with 5 μM wortmannin (Sigma) to prevent spontaneous contractions from dislodging impalements. Smooth muscle cells were impaled with glass microelectrodes filled with 2 M KCl of resistance 80–120 MΩ. Transmembrane potentials were recorded with an amplifier (Axopatch 700b; Axon Instruments) and a Digidata 1440a computer interface running pCLAMP 9.0 software.

### In vitro knockdown and contractility

Strips of GD15 or GD18 myometrium from the longitudinal layer ($2 \times 2 \times 20$ mm) were washed in sterile $Ca^{2+}$ and $Mg^{2+}$ free HBSS DMEM/F-12. Strips were placed (in triplicate) under 1.5× slack length tension in media containing 2% dextran-coated charcoal-treated foetal bovine serum with 0.5 mM 8-bromo-cAMP (Sigma), $10^{-6}$ M medroxyprogesterone acetate (Sigma) and pLenti6-cppt-CMV-mCherry-mouse 543A miRNA (Anti-Kir7.1), pLenti6-cppt-CMV-mCherry-neg miRNA (Scrambled), or pLenti6-cppt-CMV/TO-humKir7.1-IRES-mCherry-opre (+Kir7.1). Construction of lentiviral vectors is detailed in the Supplementary Methods.

On day five, strips were placed under 2mN tension in a four channel flatbed organ bath (DMT) in m-KHS solution. Isometric force was recorded on ADI Instruments LABCHART software. Activity integral was measured as the area under the time-force curve (mN/s) over a 20-min period. Contraction duration was determined as the mean duration (rise and fall to baseline) of contractions within a 20-min period. Maximum force was determined as the peak force measurement within a 20-min period. All measurements were made over the same time period for all strips.

### In vivo intrauterine pressure measurements

C57BL/6J time-mated mice were anesthetized on GD8 to GD10 by intraperitoneal injection of ketamine (100 mg/kg) plus xylazine (10 mg/kg, IP), and a PhysioTel PA-C10 transmitter (Data Sciences International) was implanted in one horn of the pregnant uterus between the uterine wall and foetal sacs. At this time, $3.7 \times 10^5$ particles/ml of pLenti6-cppt-CMV-mCherry-Neg miRNA (scrambled) or pLenti6-cppt-CMV-mCherry-miRNA mouse Kir7.1 (Anti-Kir7.1) were injected into the uterine muscle of the implanted horn. Five days later, uterine pressures were continuously measured every 10 s at 500 Hz with Dataquest A.R.T. data acquisition system version 4.31 (DSI) for 4–8 days.

### The paper explained

#### Problem

Abnormal uterine activity has profound health consequences for both mother and infant as well as health services and national wealth. Clinical conditions associated with abnormal uterine activity are preterm labour (PTL), dysfunctional labour and post-partum haemorrhage (PPH). Preterm birth is the biggest cause of neonatal mortality and morbidity. Dysfunctional labour leads to operative vaginal and abdominal delivery with its inherent maternal risks, and PPH is a major global cause of maternal morbidity and mortality accounting for around 25% of deaths in post-partum mothers in developing nations. The challenge is to understand both the heterogeneous aetiology of these conditions and to develop new methods to manipulate uterine function.

#### Results

Uterine contractions are fundamentally controlled by complex electrical signals that regulate calcium entry at the plasma membrane of the uterine myocytes. Our findings have identified Kir7.1 as a novel regulator of electrical activity of the uterus during gestation in both mice and humans. We found that the expression of Kir7.1 was regulated in pregnancy and that decreasing Kir7.1 activity *in vitro* and *in vivo* greatly increased uterine activity. Conversely, increasing Kir7.1 expression inhibited uterine contractions. We found that this effect was mediated by altering the uterine resting membrane potential and action potential. Finally, we identified novel pharmacological inhibitors of Kir7.1 that stimulated uterine contractions with greater efficacy than the currently used uterine stimulant oxytocin.

#### Impact

Our findings have identified Kir7.1 as a key regulator of uterine activity during gestation. The results of this study add to our understanding of the underlying mechanisms that control uterine activity during gestation. The development of new inhibitors to Kir7.1 may be a novel tool for manipulating uterine function for the treatment of dystocia and post-partum haemorrhage.

### Data analysis and statistics for telemetry studies

Longitudinal changes in intrauterine pressure after infection with pLenti6-cppt-CMV-mCherry-Neg miRNA (scrambled) or pLenti6-cppt-CMV-mCherry-miRNA mouse Kir7.1 (Anti-Kir7.1) were analysed by using a linear-mixed effect (LME) model implemented by PROC MIXED in SAS 9.3. Before analysis, data were first temporally aligned according to gestational days. The dependent (response) variable was taken as the hourly average intrauterine pressure. Fixed effects were treatment group (scrambled vs miRNA), gestation day and their interaction. Random intercepts were also used to incorporate mouse-specific effects. In addition, a first-order autoregressive correlation structure was used to account for the repeated measurements over time, which implies that the temporal correlation among repeated measures decays as a power function of the time lag. Residuals from initial LME analysis displayed a skewed distribution, and a square-root transformation of the response variable was found to be effective to correct the non-normality in the residuals. A few mice had negative hourly average pressures in certain hours, and 58 such observations (around 3% of the entire data set and split equally between the scrambled and miRNA group) were excluded from the analysis.

### Compound screening

Cell line generation, cell culture, and Kir7.1 automated electrophysiology assay and data analysis are all described in the Supplementary Methods. The MRCT ion channel focused compound file was selected in collaboration with the Dundee Hit Finding Unit and consisting of ~ 4,000 compounds. The set covered 119 bioactive templates from nine categories of ion channel targets and molecular weights of the screening compounds ranged between 150 and 450.

### Computational simulations

Computational simulations are described in detail the Supplementary data.

**Supplementary information** for this article is available online: http://embomolmed.embopress.org

### Acknowledgements

The authors thank the staff and patients of the University Hospitals Coventry and Warwickshire for their participation in this study. We also thank Henggui Zhang, Arun Holden and Winnie Tong for early discussions on establishing the MSM computational model. This work was supported by the Biomedical Research Unit of Warwick Medical School and University Hospitals Coventry and Warwickshire and by grants from the Medical Research Council (G0901801) to AMB; and capacity building studentship to AMB and DAR, Action Medical Research (SP4507) to AMB, MJT, AS and ST; and from the National Institute of Health (R01 HD-037831 and the March of Dimes (21-FY12-133) to SKE.

### Author contributions

HAB, DAR, SKE, JSD, MJT, CK and AMB designed the research; CM, CR, EB, SM, JA, AS, YC, JZ, DT, JJ, PW, TD, SK, DJT, PB, RC and AMB carried out the research; SQ, MV and ST, phenotyped the subjects and provided the samples; HAB, CM, NL, SKE and AMB analysed the data; and AS, HAB, MJT, SKE, JJB and AMB wrote the paper.

### Conflict of interest

The authors declare that they have no conflict of interest.

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
