## [Review Process File · EMBO Molecular Medicine]

THE INWARDLY RECTIFYING K⁺ CHANNEL KIR7.1 CONTROLS UTERINE EXCITABILITY THROUGHOUT PREGNANCY

Conor McCloskey, Cara Rada, Elizabeth Bailey, Samantha McCavera, Hugo A van den Berg, Jolene Atia, David A Rand, Anatoly Shmygol, Yi-Wah Chan, Siobhan Quenby, Jan J Brosens, Manu Vatish, Jie Zhang, Jerod S Denton, Michael J. Taggart, Catherine Kettleborough, David Tickle, Jeff Jerman, Paul Wright, Timothy Dale, Srinivasan Kanumilli, Derek J Trezise, Steve Thornton, Pamela Brown, Roberto Catalano, Nan Lin, Sarah K England & Andrew M Blanks

Corresponding author: Andrew Blanks, University of Warwick

Review timeline:

Submission date:	06 February 2014
Editorial Decision:	14 April 2014
Revision received:	13 June 2014
Accepted:	02 July 2014

Transaction Report:

Editor: Céline Carret

1st Editorial Decision

14 April 2014

Thank you for the submission of your manuscript to EMBO Molecular Medicine. We have now heard back from the three referees whom we asked to evaluate your manuscript. Although the referees find the study to be of potential interest, they also raise a number of concerns that must be addressed in a major revision of the work.

As you will see from the comments below, while referees 1 and 4 are rather positive about the study and only suggest ways to improve the manuscript for clarity, referee 2 is more critical and requests additional experiments to render the article more conclusive and compelling to a broad audience.

We would then welcome the submission of a revised version for further consideration and depending on the nature of the revisions, this may be sent back to the referees for another round of review.

Please note that it is EMBO Molecular Medicine policy to allow a single round of revision in order to avoid the delayed publication of research findings. Consequently, acceptance or rejection of the manuscript will depend on the completeness of your responses included in the next version of the manuscript.

I look forward to seeing a revised form of your manuscript as soon as possible.

***** Reviewer's comments *****

Referee #1 (Comments on Novelty/Model System):

The model systems used in this study are excellent.

Referee #1 (Remarks):

This is a remarkable study and paper, and I am very enthusiastic about the work. The manuscript is suitable for publication with modifications. My suggestions are below.

This is a very interesting and potentially a very important study and paper by a large group of excellent investigators that have combined their expertise to examine the role of a potassium channel, Kir7.1, in regulation of uterine contractility during pregnancy in models and in mice and humans. A wide range of experimental methods were used to assess the content and functional role of the Kir7.1 channel in mice and human during pregnancy. Genomic screening was used to first identify relevant potassium channels that control excitability of the myometrium. This was followed by computational modeling and evaluation of currents promoting myometrial quiescence during gestation and knockdown and overexpression studies of Kir7.1. Finally, specific Kir7.1 inhibitors, VU590 and derivatives, were used to demonstrate stimulatory effects on mouse and human myometrial contractility. The authors conclude that Kir7.1 regulates myometrial contractility during gestation by suppressing contractility and thereby aiding in maintenance of pregnancy and that a decline in Kir7.1 at term promotes labor contractions. The studies are very well done by experienced people and the manuscript is well written. Appropriate references are cited in the manuscript, including some classic older studies which are refreshing to see. As noted by the authors, the study has very significant implications for the understanding of regulation of uterine contractility during pregnancy and potential value in identification of possible treatment strategies for uterine contractility disorders, particularly postpartum hemorrhage treatment, but perhaps for other uterine contraction problems as well. Overall, I am extremely enthusiastic about the study and paper-the study group is outstanding and the study and paper are original, novel and report unique experiments that are very valuable and very important.

My specific comments are listed below.

Comments:

Abstract:

The abstract is well done but might include an additional comment in the last sentence (in bold font below). Something like the following: We conclude that Kir7.1 regulates the transition from uterine quiescence during gestation to high contractility during labor and may be the target for therapies to control uterine contractility.

Introduction:

The introduction is also well done. However, I have the following comments.

In the introduction (lines 75 to 91) the authors describe the expression of Kir7.1 that is known in other tissues. Is anything known about Kir7.1 expression in other excitable tissues? If so or if not, this should be mentioned.

Line 95--- The term hyper-excitable may not be the best terminology as hyper-excitability may mean the presence of more tonic type contractions. It may be better to simply state that removal of

the hyperpolarizing drive increases excitability and thereby increases contractility. Action potentials in a laboring myometrium have both depolarizing and hyperpolarizing components and it is not clear how a decrease in hyperpolarization currents can influence electrical events if action potentials occur more quickly (rise and fall) in laboring myometrium.

Results: Generally the results section and figures are well done and appropriate.

1. Line-106, MSM should be identified if not mentioned earlier in the paper.
2. Change is (line 142) to in.
3. Table 1-please spell out the meaning of Ct.
4. Figure 1A. --- mRNA decreases two days prior to delivery-is protein similar and if so how does one account for this before term? Is this mentioned in the discussion?
5. In most of the figures upper case letters are used but lower case letters are used in the figure legends.
6. From most of the figures it looks like inhibition of the Kir7.1 channel results in a tonic type contraction rather than contribution to channel activity that might be expected during labor when phasic activity increases. This should be discussed.
7. Figure 10 is very complex and the contribution of K⁺ channel control is lost in the figure with the addition of so many channels and pathways. This figure may be better used in a review article and replaced here with a simpler model that emphasizes how ion channels, particularly Kir7.1, play important roles in control of contractility during pregnancy and labor. One particular point on the figure is that connexins by themselves do not increase electrical coupling - this is established by assembly of connexins into gap junctions and it these contacts that are important.

Discussion:

1. Overall, the Discussion is very good. However, there are a few points that may improve it.
2. It would be useful in the Introduction and Discussion to add a very brief review of other myometrial K⁺ channels and how Kir7.1 fits into this array of channels.
3. The authors might speculate on how the change in Kir7.1 density occurs during pregnancy. Is this controlled by progesterone or receptors for progesterone or other mechanisms? Does the change in Kir.7 occur at times when other systems are also changing in the uterus? This need not be a long discussion only one or two sentences would suffice.
4. Finally, I think the biggest problem with the Discussion as seen from the Results is the lack of any comment on how a decrease in Kir7.1 channels can contribute to phasic activity of the muscle during labor, when spike activity has a rapid depolarizing element and a faster hyperpolarizing aspect. The main effect of Kir7.1 decrease and Kir7.1 inhibition seems to produce tonic contractions. If a Kir7.1 channel decrease contributes to labor contractions how is this perceived? Also, the concept that Kir7.1 could contribute to phasic activity seems contrary to the author's view that pharmacological inhibition could be mainly used for only postpartum hemorrhage where tonic contractions are needed. If Kir7.1 participates in control of labor at term would not agents that suppress it induce phasic type contractions as observed during labor? Could not agents that suppress Kir7.1 be used to augment labor, such as oxytocin? It is difficult to perceive how Kir7.1 contributes to labor contractions and it would be useful for the authors to describe this in some detail.

Referee #2 (Comments on Novelty/Model System):

I feel that the novelty of this ms is quite high. However, the ms is missing the respect for previous reports and the consistency within their data to present their idea completely.

Referee #2 (Remarks):

In this manuscript, the authors tried to identify the molecular entity for the control of quiescence of uterine contraction during gestation. By qPCR technique, they found that Kir7.1 is the only K⁺ channel enriched in myometrial smooth muscles. The expressions of Kir7.1 at levels of mRNA and protein were increased during gestation and decreased after mid-gestation. They measured the current sensitive to VU590, a Kir7.1 specific blocker, from the myometrial cells. Disruption of the protein expression by miRNA of Kir7.1 or the administration of a known Kir7.1-blocker and a new blocker led to the elevation of V_m and the increase in the tone contraction of myometrial cells. Conversely, over-expression of Kir7.1 lowered the V_m and reduced the activity of myometrial cells.

Furthermore, they performed the simulation study to test how the current density of Kir7.1 associates with the Ca²⁺ influx. These studies are comprehensive and mostly cover their idea that Kir7.1 controls uterine excitability during pregnancy. The followings are comments that I believe to strengthen their idea.

Major)

1. Previously KATP channels and two-pore domain K⁺ channels have been proposed to be K⁺ currents to control the excitability of myometrium. However, the authors just only provided the primer sequence information without presenting the expression level of other K⁺ channels. Furthermore, there is no discussion about these channels. Although the expression of Kir7.1 is critically regulated during pregnancy, if the basal channel expression level of other K⁺ channels is high, these channels would effectively control the contraction and be potential therapeutic targets. It had better provide experimental data and discuss the ties with other K⁺ channels in the control of quiescence of uterine contraction.
2. The simulation study would be problematic. In the supplementary mathematical methods, the authors provide the expression of IK1 in myometrium cell model without any interpretations. However, no module corresponding to the IK1 was built in the final model. Although this might relate to the comment mentioned above, how much the inward K⁺ currents are expressing in the myometrium cells? Furthermore, how do we compare their results with the biological phenomenon? Probably the simulation for the V_m in Fig. 3 would correspond to those described in Fig. 5. In Fig. 5, they showed the change in the current density of Kir7.1 dramatically alters the duration, configuration and V_m of the action potential in myometrical strips. However, the simulation failed to reproduce such features (Fig. 3).
3. Concerning about the mechanistic insights into gestation-dependent contraction, the low current density of Kir7.1 might increase the sensitivity to channel blockers and lead to the contraction before deliver (GD18).

Minor)

1. Are there any better immunostaining to provide the membrane distribution of Kir7.1 in Fig. 1?
2. A membrane potential measured this current density in Fig. 2c is suggested to be -100 mV in the legend, but -150 mV in the result section.
3. Since the recording condition of electrophysiological property of myometric cells is unclear, it is difficult to judge whether the VU590-sensitive inward current is a K⁺ current.
4. The label, "anti-Kir7.1" in Fig. 4 and 5 is confusable.
5. Why did the authors apply oxytocin with VU590 in Fig. 7B? Was Oxytocin required to produce this phenomenon?

Referee #4 (Remarks):

This manuscript uses an impressive combination of molecular profiling, immunochemistry, viral-mediated knockdown/expression, electrophysiology and in vivo telemetric recordings to provide strong evidence that Kir7.1 is expressed in mouse and human myometrium where it provides a brake on cellular excitability and myometrial contractility. The expression of Kir7.1 changes throughout the gestational period, such that levels are relatively low (but not absent) at the time of parturition. An experimental decrease in channel expression (by using knockdown) and/or channel activity (by using pharmacological block) increased myometrial cell electrogenic activity, strip tension and intrauterine pressure; the pharmacological effect of Kir7.1 blockers appeared to exceed or enhance actions of oxytocin on myometrial contractility, increasing contraction duration, and providing some rationale for the suggestion that such blockers could find use in the uterine atony that often leads to postpartum hemorrhage.

Overall, this is a very strong set of studies that makes a convincing case that Kir7.1 channels play an important role in regulating myometrial cell excitability and contractility. It will provide a compelling rationale for exploring Kir7.1 blockers for controlling uterine function.

A few concerns should be addressed to clarify some of the results.

1. For the experiments on human myometrial strips depicted in Fig. 7B, it is not clear how one can discern the independent effect of VU590 since it appears to be co-applied with oxytocin. What is the effect of OT alone vs. VU590 alone?

2. For the experiments on mouse myometrium in Fig.8B, the scale makes it hard to see if oxytocin had any effect on its own. What were those effects? Also, is the implication that the actions of oxytocin and VU590 are synergistic. It might be good to state this explicitly. In this respect, some Kir channels are inhibited by PIP2 depletion, which might be expected following OT receptor stimulation. Is this the case for Kir7.1? Could this be a mechanism for synergy?

3. For the experiments on human myometrium in Fig.8C, what accounts for the absence of any effect of VU590 alone at the 100 μ M concentration? Were effects on contraction duration in Fig. 8D from the experiment with OT or without OT? (Please also state in the legend that these strips were taken from the uterus at "term".

4. Please clarify the determination of activity integral. For example, in Fig. 4B, there appears to be an increase in basal tone (the starting point on the Y-axis is at ~ 1.3 vs ~ 0 in the scrambled (4A) or the overexpression (4C) examples. Does this basal tone also get included in the integral? or is the integral determined relative to the starting level?

Also, what is the contribution of frequency to the measure of activity integral? The differences in frequency associated with Kir7.1 expression appear to be pronounced in the cell activity levels (e.g., in Fig. 5).

5. The text appears to suggest that the effect of Kir7.1 knockdown on intrauterine pressure is enhanced at GD15 relative to GD18 (see l. 155) but there is no indication of statistical significance in the relevant Fig. 6 histogram plot (compare black bars.) In addition, despite different levels of Kir7.1 expression and activity at GD15 and GD18 (Figs. 1 & 2), there is no apparent difference in baseline intrauterine pressure (compare grey bars in Fig. 6). Does this mean that some other mechanisms are offsetting the loss of Kir7.1 at GD18 to maintain the reduced intrauterine pressure?

6. Which library of known ion channel inhibitors was screened?

7. line 92: "In this study, we demonstrate the crucial importance of Kir7.1 in maintaining pregnancy..." is not strictly accurate.

8. Introduction: Some mutations in Kir7.1 are described in various species. Were there any relevant reproductive deficiencies noted in these animals?

1st Revision - authors' response

13 June 2014

Response to Reviewers McCloskey *et al.*

Referee #1:

We would like to thank the reviewer for their strong support for our work and their insightful comments. We address the individual points below:

1) *Abstract:*

"The abstract is well done...uterine contractility"

Reply:

We agree with the Reviewer and this has now been changed.

2) *"In the introduction (lines 75 to 91) the authors describe the expression of Kir7.1 that is known in other tissues. Is anything known about Kir7.1 expression in other excitable tissues?"*

Reply:

Yes, we mentioned that Kir7.1 is expressed in neurons where it is believed to control resting membrane potential and have now emphasized this point.

3) *"Line 95--- The term hyper-excitable may not be the best terminology as hyper-excitability may mean the presence of more tonic type contractions. It may be better to simply state that removal of the hyperpolarizing drive increases excitability and thereby increases contractility. Action potentials in a laboring myometrium have both depolarizing and hyperpolarizing components and it is not clear how a decrease in hyperpolarization currents can influence electrical events if action potentials occur more quickly (rise and fall) in laboring myometrium."*

Reply:

We agree this may be confusing. We have changed 'hyper-excitable' to 'more excitable'.

4) *1. Line-106, MSM should be identified if not mentioned earlier in the paper.*

Reply:

MSM is defined on line 58 of the Introduction immediately after it is first used. It is also identified as a non-standard abbreviation on page 2.

5) *2. Change is (line 142) to in.*

Reply:

Thank you. This has been changed.

6) *3. Table 1-please spell out the meaning of ΔCt .*

Reply:

The method for this calculation and its meaning are described in detail in E methods.

7) *4. Figure 1A. --- mRNA decreases two days prior to delivery-is protein similar and if so how does one account for this before term? Is this mentioned in the discussion?*

Reply:

Yes. As depicted in Figure 2, the functional current is decreased on GD18 when compared to GD15. Our hypothesis is that the decrease in Kir7.1 current would need to occur prior to term to allow the myometrium to become susceptible to stimulation once changes in other contraction associated proteins have taken place.

8) *5. In most of the figures upper case letters are used but lower case letters are used in the figure legends.*

Reply:

Thank you; we have changed all legends to upper case.

9) 6. From most of the figures it looks like inhibition of the Kir7.1 channel results in a tonic type contraction rather than contribution to channel activity that might be expected during labor when phasic activity increases. This should be discussed.

Reply:

Inhibition of Kir7.1 with higher doses of VU590 does indeed lead to tonic type contractions but lower doses do not. The knockdown experiments, depicted in Figures 4 and 5, demonstrate the contractions remain phasic. This effect is predicted from the two components through which Kir7.1 contributes to the action potential: (i) the resting membrane potential and (ii) the general excitability during the depolarized phase of the phasic contraction. This is addressed in the Discussion, lines 237 and 247. Set in a physiological context, the reduction but not complete removal of Kir7.1 during gestation results in a more depolarized membrane potential, and a more excitable depolarized phase whilst maintaining the phasic contraction. This is explained in more detail in reply 13 below and is explored quantitatively in the simulations now provided in Figure E6.

10) 7. Figure 10 is very complexconnexins into gap junctions and it these contacts that are important.

Reply:

Thank you for this suggestion. We feel that it is very important to set the effect of modulation of the action potential within the context of the overall changes that are occurring during the end of pregnancy. This is because most work in this area focuses on hormonal/inflammatory regulation of contractions without considering the biophysical components. We therefore see this as an opportunity to educate and inform. Accordingly, we have kept the diagram but have emphasised the excitation-contraction coupling component of the control process. In addition to this, the addition of Figure E6 (with extra simulation data) now explain in detail the effect changing the Kir7.1 conductance has on other ion channels during the myometrial action potential. When taken together, the two diagrams give a very comprehensive explanation of where Kir7.1 sits within the overall process of phasic uterine contractions. Finally, we agree entirely with the Reviewer with respect to gap junctions and clarified this in the legend to prevent any misconceptions.

11) It would be useful in the Introduction and Discussion to add a very brief review of other myometrial K⁺ channels and how Kir7.1 fits into this array of channels.

Reply:

Reviewer 2 also requested further background information on K⁺ channels in the myometrium together with some functional context. We have included this in the revised Introduction (lines 76-82) and kept the Discussion focused on inwardly rectifying potassium channels. To ascertain the physiological context, and assess how the behavior of Kir7.1 relates to the other conductances, we include additional simulation data in Figure E6 explaining the role of each conductance and how it is affected by changes in Kir7.1 channel density. This simulation investigates, in a quantitative way, how interlinked the different ion channels are in their behavior. This is important because the effect of changing an individual conductance within a nonlinear system, such as the action potential, can be very difficult to predict intuitively. This was our primary motivation for undertaking computer modeling.

12) 3. The authors might speculate on how the change in Kir7.1 density occurs during pregnancy. Is this controlled by progesterone or receptors for progesterone or other mechanisms? Does the change in Kir.7 occur at times when other systems are also changing in the uterus? This need not be a long discussion only one or two sentences would suffice.

Reply:

Thank-you for the suggestion. It is always difficult to speculate in the absence of evidence but it would seem unlikely that progesterone regulates Kir7.1 as the fall in its expression in the mouse uterus precedes progesterone withdrawal by two days. This raises the intriguing possibility of a gestation-dependent but progesterone-independent change altering uterine excitability during pregnancy. We have now included a sentence on this in the Discussion (lines 239-241).

13) 4. Finally, I think the biggest problem with the Discussion as seen from the Results is the lack of any comment on how a decrease in Kir7.1 channels can contribute to phasic activity of the muscle during labor, when spike activity has a rapid depolarizing element and a faster hyperpolarizing aspect. The main effect of Kir7.1 decrease and Kir7.1 inhibition seems to produce tonic contractions. If a Kir7.1 channel decrease contributes to labor contractions how is this perceived? Also, the concept that Kir7.1 could contribute to phasic activity seems contrary to the author's view that pharmacological inhibition could be mainly used for only postpartum hemorrhage where tonic contractions are needed. If Kir7.1 participates in control of labor at term would not agents that suppress it induce phasic type contractions as observed during labor? Could not agents that suppress Kir7.1 be used to augment labor, such as oxytocin? It is difficult to perceive how Kir7.1 contributes to labor contractions and it would be useful for the authors to describe this in some detail.

Reply:

This is quite straightforward. The biophysics of Kir7.1 means that its open state probability is high at the resting membrane potential and during the depolarized phase of the action potential. Within physiological trans-membrane voltages and gradients this means that there is a consistent current passing through the channel at all times. Furthermore as the membrane potential deviates away from the reversal potential for potassium, the driving force increases. Thus a decrease in Kir7.1 has two main effects. Firstly, to alter resting membrane potential and, secondly, to stress the repolarizing drive of the action potential as the Kir7.1 current decreases. This is illustrated in detail in the simulation depicted in the Figure E6 and is covered in the Discussion, lines 246-256. The general principle of repolarization reserve is discussed in detail in Fink *et al* in the References.

Referee #2:

We would like to thank the reviewer for their positive comments and address their specific concerns below:

Major)

1. Previously KATP channels and two-pore domain K⁺ channels have been proposed to be K⁺ currents to control the excitability of myometrium. However, the authors just only provided the primer sequence information without presenting the expression level of other K⁺ channels. Furthermore, there is no discussion about these channels. Although the expression of Kir7.1 is critically regulated during pregnancy, if the basal channel expression level of other K⁺ channels is high, these channels would effectively control the contraction and be potential therapeutic targets. It had better provide experimental data and discuss the ties with other K⁺ channels in the control of quiescence of uterine contraction.

Reply:

The Reviewer is correct in stating that other potassium channels are very important in controlling the myometrial action potential. Indeed, our model is the most comprehensive produced to date and includes accurate biophysics for all possible species expressed within the myometrium to quantitatively assess their function. It is important to clarify that the expression data in the table refer to genes that were differentially expressed between myometrial smooth muscle and the vasculature. It is our reasoning that a drug that targets channels enriched in myometrial smooth muscle cells over the vasculature has a greater chance of not having undesirable cardiovascular side effects (as K_{ATP} might). This does not mean that genes that are not on this list but expressed in myometrial smooth muscle are not important. On the contrary, many of them are and are in the model because they are required for the parsimonious fit solution. We now touch on other known ion channels in the Introduction and included simulations to quantitatively assess the effect of altering Kir7.1 on other myometrial conductances. A crucially important point here is that our

parsimonious fit approach is the best solution of the biophysics to the physiological observation on which it was trained (in this case recorded spontaneous action potentials). In reality a physiological system such as uterine smooth muscle experiences many different situations either during homeostasis or under environmental challenges such as low energy levels for example. Under such conditions, and to use the example given by the reviewer of K_{ATP} , one might expect the lower ATP levels to open a population of closed channels and slow the myometrium. Such an explanation fits well with observed experimental data where spontaneously contracting myometrium is unaffected by Glibenclamide (i.e. there is no open population to close) but responds well to Pinacidil (i.e. there is a large closed population to open (Piper et al. Br J. Pharmacol (1990),101,901-907)). Though of interest physiologically such circumstances are not the subject of this study or experimental design. Put another way, our modeling approach suggested that only those conductances expressed at the densities predicted were required to produce the spontaneous action potentials used to train the model. This in no way excludes other conductances being important under different physiological conditions. For the purposes of this study we used the model to interrogate the effect of changing Kir7.1 densities on the myometrium and tested this hypothesis empirically.

2. In the supplementary mathematical methods, the authors provide the expression of IK1 in myometrium cell model without any interpretations. However, no module corresponding to the IK1 was built in the final model. Although this might relate to the comment mentioned above, how much the inward K+ currents are expressing in the myometrium cells?

Reply:

We thank the Reviewer for identifying this error. The IK1 in the diagram is a transcribing error from a paper and actually refers to SK4. This has now been changed. We take “how much the inward K+ currents are expressing in the myometrium cells?” to mean how are the channel densities determined? The channel densities within our model are calculated based on our fitted method to create the most parsimonious solution to create the full myometrial action potential as determined by empirical observation. The methods for this are given in full detail in E data.

Furthermore, how do we compare their results with the biological phenomenon? Probably the simulation for the Vm in Fig. 3 would correspond to those described in Fig. 5. In Fig. 5, they showed the change in the current density of Kir7.1 dramatically alters the duration, configuration and Vm of the action potential in myometrical strips. However, the simulation failed to reproduce such features (Fig. 3).

Reply:

It is certainly the case that inhibiting all or most of the current appears to cause long lasting contractions but as can be seen in Figure 5 specific knock-down of Kir7.1 expression leads to phasic depolarisations that are individually the same duration but fail to remain at resting membrane potential initiating repeated events. This is not sufficient time for the contractile machinery or calcium extrusion mechanisms to relax the muscle and contract again. The kind of disordered contraction this leads to can be seen in Figure 4B. The simulation is intended to interrogate one single depolarization event in detail and replicates the above phenomena (note the difference in plateau potential in 5C and 5D). The situation with higher concentrations of our compounds is a more extreme phenotype than this with prolonged contraction durations and depolarization as depicted in Figure 7. The difference in these responses, which emerges at higher concentrations of the compound, are not yet fully understood but may be due to non-specific inhibition of other potassium channels. This work is ongoing but does not have any bearing on the observations made in the knockdown experiments *in vitro* and *in vivo* and simulations; nor does it necessarily preclude the use of inhibitors as a therapy since compounds that target a single channel alone at therapeutic doses are extraordinarily rare.

3. Concerning about the mechanistic insights into gestation-dependent contraction, the low current density of Kir7.1 might increase the sensitivity to channel blockers and lead to the contraction before deliver (GD18).

Reply:

We are unsure what the Reviewer means here. In our experiments, myometrial strips were more sensitive to block by VU590 (Figure 8a) in mid gestation when the expression of Kir7.1 was greatest (Figures 1 & 2). Under normal physiological conditions, as the gestation proceeds to term, the uterus undergoes other changes such as increased susceptibility to OT stimulation by up regulation of the OTR. Thus once the Kir7.1 inhibition is removed and the OTR up regulated, the uterus is capable of a more potent response (see Figure 8b).

Minor)

1. Are there any better immunostaining to provide the membrane distribution of Kir7.1 in Fig. 1?

Reply:

We feel the immunostaining in Figure 1 illustrates well that the expression of Kir7.1 is confined to myometrial but not vascular smooth muscle. Membrane expression is confirmed in the functional electrophysiology experiments undertaken in the whole cell perforated patch configuration whilst the western blots on different tissues suggest antibody specificity.

2. A membrane potential measured this current density in Fig. 2c is suggested to be -100 mV in the legend, but -150 mV in the result section.

Reply:

Thank you for identifying this error in the Figure legend. It is now changed to -150 mV.

3. Since the recording condition of electrophysiological property of myometric cells is unclear, it is difficult to judge whether the VU590-sensitive inward current is a K⁺ current.

Reply:

Pipette recording solution details have now been added to the Methods section. Under these conditions, the current recorded reversed at the reversal potential for potassium (-80mV).

4. The label, "anti-Kir7.1" in Fig. 4 and 5 is confusable.

Reply:

We are unsure what the Reviewer means but the term is defined for the reader in the Legend.

5. Why did the authors apply oxytocin with VU590 in Fig. 7B? Was Oxytocin required to produce this phenomenon?

Reply:

We applied VU590 alone and in combination with oxytocin as can be seen in detail in Figure 8A, B, C & D. OT is not required for the VU590 response but the response to OT is potentiated by co-administration of VU590. We applied oxytocin for two reasons. Firstly, to provide a direct comparison to the currently used agent for the treatment of postpartum haemorrhage and, secondly, because oxytocin is one of the naturally occurring stimulants of the uterus at term.

Referee #4:

We would like to thank the Reviewer for his/her strong support for our work and the insightful comments. We address the individual points below:

1) For the experiments on human myometrial strips depicted in Fig. 7B, it is not clear how one can discern the independent effect of VU590 since it appears to be co-applied with oxytocin. What is the effect of OT alone vs. VU590 alone?

The detailed breakdown and comparison of the effect of VU590 alone compared with combined oxytocin dose is depicted in Figure 8. OT alone is depicted in Figure 8B and VU590 alone in figure 8A. Murine myometrium is depicted in panels 8A and 8B and human myometrium in panels 8C and 8D.

2). For the experiments on mouse myometrium in Fig.8B, the scale makes it hard to see if oxytocin had any effect on its own. What were those effects?

The effects of OT, alone or in combination, are depicted in Figure 8B and described in the text on lines 178-179. "The effect of VU590 on contractile force in samples taken from GD15 and GD18 mice was dependent on dose and gestation (Fig. 8a). When used in combination with oxytocin, VU590 increased the activity integral by 172 ± 14 fold and 90 ± 42 fold on GD15 and GD18 respectively, as compared to 4 ± 2 fold and 8 ± 3 fold for OXT treatment alone (Fig. 8b)."

Also, is the implication that the actions of oxytocin and VU590 are synergistic. It might be good to state this explicitly. In this respect, some Kir channels are inhibited by PIP2 depletion, which might be expected following OT receptor stimulation. Is this the case for Kir7.1? Could this be a mechanism for synergy?

Thank you for this insightful comment. It is indeed possible that the actions are synergistic via a PIP2 mechanism. There is evidence that Kir7.1 is sensitive to PIP2. We have now mentioned this in the Discussion (lines 217-219).

3. For the experiments on human myometrium in Fig.8C, what accounts for the absence of any effect of VU590 alone at the 100 μ M concentration?

Excellent observation. The 100 μ M dose causes a profound depolarization, which ensures that the depolarized membrane potential is no longer within the window current for calcium entry. This leads to the generation of very little force.

Were effects on contraction duration in Fig. 8D from the experiment with OT or without OT?

These are without OT administration as indicated in the text (line 179-181) as well as in the legend of Figure 8.

(Please also state in the legend that these strips were taken from the uterus at "term".

Reply:

We have added this but it is important to clarify that all human samples in this study were taken at term.

4. Please clarify the determination of activity integral. For example, in Fig. 4B, there appears to be an increase in basal tone (the starting point on the Y-axis is at ~ 1.3 vs ~ 0 in the scrambled (4A) or the overexpression (4C) examples. Does this basal tone also get included in the integral? or is the integral determined relative to the starting level?

Reply:

The method has now been included in the methods section. The basal tone is not included in the analysis. The integrating algorithm takes the pre-contraction tone as an averaged baseline and integrates above this line until the contraction relaxes back to this value.

Also, what is the contribution of frequency to the measure of activity integral? The differences in frequency associated with Kir7.1 expression appear to be pronounced in the cell activity levels (e.g., in Fig. 5).

Reply:

Frequency of contraction is part of the activity integral measurement because this is measured over a 20-minute period. However, as can be seen in Figure 4, the main component of the increased activity integral upon knockdown of Kir7.1 was contraction duration. The contraction duration was increased on average because the membrane potential in the knockdown strips fluctuated rapidly between the resting and depolarized state (Figure 5B), leaving insufficient time for calcium extrusion and relaxation.

5. The text appears to suggest that the effect of Kir7.1 knockdown on intrauterine pressure is enhanced at GD15 relative to GD18 (see l. 155) but there is no indication of statistical significance in the relevant Fig. 6 histogram plot (compare black bars.) In addition, despite different levels of Kir7.1 expression and activity at GD15 and GD18 (Figs. 1 & 2), there is no apparent difference in baseline intrauterine pressure (compare grey bars in Fig. 6). Does this mean that some other mechanisms are offsetting the loss of Kir7.1 at GD18 to maintain the reduced intrauterine pressure?

Reply:

This is an interesting question and we thank the Reviewer for the suggestion. We originally used a linear-mixed effect (LME) model implemented by PROC MIXED in SAS 9.3. Before analysis, data were first temporally aligned according to gestational days. The dependent (response) variable was taken as the hourly average intrauterine pressure while fixed effects were treatment group (scrambled vs miRNA), gestation day, and their interaction. To investigate the Reviewer's question about the baseline and gestational age we re-analysed the data to test whether there was a significant change in intrauterine pressure as a function of gestation and independent of treatment. We found that there was no statistically significant difference in either treatment group as a function of gestation. By contrast mice in which Kir7.1 was knocked down had significantly increased intrauterine pressure when compared to mice injected with scrambled miRNA lentivirus from GD13 to GD18. These data are now presented in a new Figure 6. It does appear that the two groups converge at the end of gestation and so the Reviewers' insight about a gestation dependent offset may well be correct. This is a concept that may well be worth exploring further in the future.

6. Which library of known ion channel inhibitors was screened?

Reply:

We used an MRCT ion channel focused compound file consisting of ~4,000 compounds. This set was selected in collaboration with the Dundee Hit Finding Unit. The set covers 119 bioactive templates from 9 categories of ion channel targets and molecular weights of the screening compounds range between 150 and 450.

7. line 92: "In this study, we demonstrate the crucial importance of Kir7.1 in maintaining pregnancy..." is not strictly accurate.

Reply:

We have removed "in maintaining pregnancy".

8. Introduction: Some mutations in Kir7.1 are described in various species. Were there any relevant reproductive deficiencies noted in these animals?

The mutation referred to was described in Zebrafish, where it contributes to a malfunction in stripe patterning during development. In this study, no reproductive defect was reported.

2nd Editorial decision

02 July 2014

Please find enclosed the final reports on your manuscript and thank you for getting back to me quickly with the requested modifications. I am happy to inform you that your manuscript is accepted for publication and is now being sent to our publisher to be included in the next available issue of EMBO Molecular Medicine.

Congratulations on your interesting work.

***** Reviewer's comments *****

Referee #1 (Comments on Novelty/Model System):

This is a very interesting and potentially important paper on control of myometrial contractility during pregnancy. The authors have revised the manuscript and addressed all the reviewer's comments.

Referee #1 (Remarks):

The manuscript has been thoroughly revised based upon two reviewer's suggestions. I do not think that additional revisions are necessary.

Referee #2 (Comments on Novelty/Model System):

The authors have responded thoroughly to the reviewers comments. I have been satisfied with their responses as well as the revision of the manuscript. I believe the revised paper will serve as a valuable contribution to the treatment of clinical conditions deficient for uterine activity.

Referee #2 (Remarks):

The authors have responded thoroughly to the reviewers comments. I have been satisfied with their responses as well as the revision of the manuscript. I believe the revised paper will serve as a valuable contribution to the treatment of clinical conditions deficient for uterine activity.

Referee #4 (Remarks):

I have no further comments on the manuscript -- very nice work.